# An Optimization of Oregano, Thyme, and Lemongrass Essential Oil Blend to Simultaneous Inactivation of Relevant Foodborne Pathogens by Simplex–Centroid Mixture Design

**DOI:** 10.3390/antibiotics11111572

**Published:** 2022-11-08

**Authors:** Luiz Torres Neto, Maria Lúcia Guerra Monteiro, Maxsueli Aparecida Moura Machado, Diego Galvan, Carlos Adam Conte Junior

**Affiliations:** 1Center for Food Analysis (NAL), Technological Development Support Laboratory (LADETEC), Cidade Universitária, Rio de Janeiro 21941-598, RJ, Brazil; 2Laboratory of Advanced Analysis in Biochemistry and Molecular Biology (LAABBM), Department of Biochemistry, Federal University of Rio de Janeiro (UFRJ), Cidade Universitária, Rio de Janeiro 21941-909, RJ, Brazil; 3Graduate Program in Food Science (PPGCAL), Institute of Chemistry (IQ), Federal University of Rio de Janeiro (UFRJ), Cidade Universitária, Rio de Janeiro 21941-909, RJ, Brazil; 4Graduate Program in Veterinary Hygiene (PPGHV), Faculty of Veterinary Medicine, Fluminense Federal University (UFF), Vital Brazil Filho, Niterói 24220-000, RJ, Brazil; 5Institute of Chemistry, Federal University of Mato Grosso do Sul (UFMS), Campo Grande 79070-900, MS, Brazil; 6Graduate Program in Sanitary Surveillance (PPGVS), National Institute of Health Quality Control (INCQS), Oswaldo Cruz Foundation (FIOCRUZ), Rio de Janeiro 21040-900, RJ, Brazil

**Keywords:** natural antimicrobials, volatile oils, bioactive compounds, minimum inhibitory concentration, minimum bactericidal concentration, desirability function

## Abstract

(1) Background: This study aimed to use the simplex–centroid mixture design methodology coupled with a microdilution assay to predict optimal essential oil (EO) formulations against three potential foodborne pathogens simultaneously through the desirability (D) function. (2) Methods: Oregano (ORE; *Origanum vulgare*), thyme (THY; *Thymus vulgaris*), and lemongrass (LG; *Cymbopogon citratus*) and their blends were evaluated concerning minimum inhibitory concentration (MIC) and minimum bactericidal concentration (MBC) for *Salmonella enterica* serotype Enteritidis, *Escherichia coli* and *Staphylococcus aureus.* (3) Results: THY combined with ORE or LG were the most promising EO formulations in inhibiting and killing each bacterium separately. Regarding the simultaneous effect, the optimal proportion for maximum inhibition was composed of 75% ORE, 15% THY, and 10% LG, while for maximum inactivation was 50% ORE, 40% THY, and 10% LG. (4) Conclusion: The multiresponse optimization allowed identifying an EO blend to simultaneously control three potential foodborne pathogens. This first report could be a helpful natural and green alternative for the industry to produce safer food products and mitigate public health risks.

## 1. Introduction

Foodborne pathogens are a global public health issue with over 600 million cases per year resulting in concern towards morbidity, hospitalizations, and mortality, adding 420,000 deaths, and US$ 110 billion lost each year worldwide, requiring alternatives to produce safer food products [1,2,3,4]. In this context, *Salmonella enterica* serotype Enteritidis, *Escherichia coli*, and *Staphylococcus aureus* are responsible for several cases of food outbreaks worldwide [4,5,6]. They are the most common foodborne pathogens affecting the health of millions of people and adding up to an annual economic loss of billions of dollars [7,8,9,10]. Moreover, the infectious diseases caused by them include several harsh symptoms such as stomach cramps, diarrhea, fever, nausea, and/or vomiting, and even life-threatening conditions [11].

Essential oils (EOs) are rich in a broad biological spectrum of compounds such as sesquiterpenes, monoterpenes, aldehydes, alcohols, esters, ketones, polyphenols, and flavonoids, including other classes, which result in a broad biological activity including antimicrobial, antifungal, insecticide, anti-inflammatory, antioxidant, anticarcinogenic, and antiviral [12,13]. Therefore, these oils have been successfully applied in different industrial sectors such as food, cosmetic, pharmaceutical, and agricultural industries [14].

EOs are considered safe and eco-friendly plant-based antimicrobial alternatives to control foodborne pathogens and other microorganisms, including those drug-resistant ones [15,16]. Previous studies have already reported the potential of EOs against *S. enteritidis*, *E. coli*, and *S. aureus*, maintaining food quality and safety [17,18]. However, the EO concentrations needed to achieve antimicrobial effectiveness have generally led to adverse sensory impacts in foods due to their intense aroma. In this way, blended formulations have recently been investigated to reach the antimicrobial goal using lower EO concentrations since the effect of EO mixtures could be boosted by interactions of different functional groups [19,20,21]. Furthermore, EO blends have been drawing attention as a promising green technology to decrease antibiotic resistance among bacteria, adverse effects on human health, and environmental impacts from synthetic compounds due to their toxicity and slow degradation periods [22,23]. Nevertheless, the wide variety of EOs and foodborne pathogens makes it a stiff challenge.

The Mixture Designs (MDs) can assist in achieving optimized EO blends. In this experimental design, two or more components are combined in different proportions, and the results can be modeled and predicted mathematically and graphically [24,25]. One approach is to apply these models combined with desirability functions to optimize multiple responses simultaneously with a reduced number of experiments but with high quality and low cost [26]. The MDs were already used to maximize or optimize the antimicrobial activity of essential oil blends against different pathogens [16,18,21,27]. However, despite the great potential of this design in evaluating EO blends, this yet is underused. Moreover, these studies evaluate neither the combination of oregano, thyme, and lemongrass nor the simultaneous inhibition and inactivation of foodborne pathogens.

In this context, this study aimed to achieve optimized formulations containing oregano, thyme, and lemongrass EOs for individual and simultaneous inhibition and inactivation of *S. enteritidis*, *E. coli*, and *S. aureus* through an augmented simplex–centroid mixture design, attempting safer food products and public health improvement.

## 2. Results

### 2.1. Chemical Composition of Essential oils

The complete profile of compounds in EOs is available in Appendix A. Oregano EO exhibited a high concentration of carvacrol (70.3%), followed by other compounds such as p-cymene (10.4%), γ-terpinene (4.8%), (E)-β-caryophyllene (4.7%), linalool (2.2%), and myrcene (1.6%). The thyme EO was also predominantly composed of phenolic compounds, which were thymol (31.2%) and carvacrol (25.5%), followed by p-cymene (21.7%), linalool (6%), limonene (3.4%) and borneol (3.4%). Concerning lemongrass EO, its composition was rich in aldehydes (45.5% geranial and 33.7% neral) and other compounds, such as geraniol (4.5%), geranyl acetate (1.5%), citronellal (1.3%), and citronellol (1.1%). Similar chemical composition for oregano [28], thyme [29,30], and lemongrass [31,32] EOs have been previously reported in the literature.

### 2.2. Single and Combined Antimicrobial Effects through Mixture Design (MD)

Table 1 shows the ANOVA of the regression models for each bacterium concerning MIC and MBC with the corresponding *F*-values and *p*-values as well as the corresponding *R*^2^ and adjusted *R*^2^ values demonstrating the quality of the selected quadratic models to data adjustment, except for *S. aureus*. As all least-square regression models were previously evaluated for better data adjustment, this *S. aureus* strain (ATCC 14458) was more complex to model. However, it should be noted that most individual models were significant and did not show a lack of fit in most cases. In addition to these factors, the critical values were experimentally validated, as described in the following sections.

The effects obtained by the MD allowed identifying the different bacteriostatic (MIC) and bactericidal (MBC) effects of single and blended EOs (Table 2). Concerning single EOs (100% of ORE, THY, or LG), the individual effects of ORE^1^ were not significant in *E. coli*; in contrast, THY^1^ and LG^1^ demonstrated a significant effect for inhibiting and killing *E. coli*, mainly THY^1^, which exhibited a lower coefficient than LG^1^, (see Table 2). LG^1^ was the only one among the single EOs that showed a significant coefficient regarding bacteriostatic and bactericidal effects for *S. enteritidis*. THY^1^ and LG^1^ demonstrated the ability to inhibit *S. aureus*, mainly THY^1,^ due to the lowest coefficient (Table 2). Likewise, for *E. coli*, no significant coefficient effect was observed for ORE^1^ on *S. aureus* and *S. enteritidis* inhibition for MIC (Table 2).

Regarding the mixtures of EOs (proportion 50%:50%), the promising blended EOs were those with negative coefficient values, indicating an increased antibacterial effect by acting additively or synergistically (Table 2). Otherwise, a positive coefficient value suggests an antagonist effect [33]. In this way, a significant effect was observed in the ORE^0.5^ + LG^0.5^ and THY^0.5^ + LG^0.5^ for inhibiting and killing *E. coli* (Table 2), especially THY^0.5^ + LG^0.5^, due to the lowest coefficients, revealing an interesting combination against this bacterium. In contrast, although ORE^0.5^ + THY^0.5^ have exhibited a negative coefficient value, its bacteriostatic and bactericidal effects were not significant for *E. coli* (Table 2). The same was observed for *S. enteritidis*, wherein ORE^0.5^ + THY^0.5^ was not effective in inhibiting this bacterium, and none of the three blended EOs (50%:50%) in killing it (MBC) (Table 2). Otherwise, ORE^0.5^ + LG^0.5^ and THY^0.5^ + LG^0.5^ demonstrated a significant effect on *S. enteritidis* inhibition (MIC). Among these two blended EOs, ORE^0.5^ + LG^0.5^ had a lower coefficient than THY^0.5^ + LG^0.5^, indicating that this mixture of oregano and lemongrass has a greater potential to inhibit *S. enteritidis* (Table 2). Regarding *S. aureus*, all blended EOs were not significant for both the MIC and MBC values. Moreover, THY^0.5^ + LG^0.5^ showed a positive coefficient for the MBC values of *S. aureus*, while the lowest ones were against *E. coli* and *S. enteritidis* (Table 2).

The combination of ORE^0.5^ and THY^0.5^ was the only EO blend not significant for the MIC and MBC values of the three bacteria tested. In addition, this EO blend had positive coefficients for the MIC and MBC values of *S. enteritidis*. On the contrary, when combined with LG, ORE, and THY, there is evidence of a synergistic or complementary potential.

The treatments obtained through MD were experimentally evaluated for all bacteria strains. From these data, equations and 3D graphics with the different combinations of the three EOs were generated (Figure 1), which allowed identifying through mathematical models the EO proportions for potentially achieving the lowest MIC and MBC values for all bacteria strains (darkest green regions). The optimal proportions to reach the best inhibitory and bactericidal effect for *E. coli* were 100% ORE (Figure 1A,D). For *S. aureus*, the ideal mixtures observed were 50% ORE and 50% THY to achieve the maximum inhibition (Figure 1B), and 50% ORE and 50% LG or THY (saddle surfaces) to reach the highest bactericidal effect (Figure 1E). For *S. enteritidis*, the optimal proportions to achieve the best inhibitory and bactericidal effect were 75% ORE, 10.7% THY, and 14.3% LG, and 100% ORE, respectively (Figure 1C,F).

### 2.3. Mixture Optimization and Validation

The desirability function (D) (Section 5.4.4) allowed the identification of the optimized theoretical mixtures of ORE, THY, and LG to maximize the bacteriostatic (MIC; Figure 2) and bactericidal (MBC; Figure 3) actions against *E. coli*, *S. aureus*, and *S. enteritidis* simultaneously. The optimal EO blend for inhibiting bacteria consisted of 75% ORE, 15% THY, and 10% LG, which minimized the MIC values to 0.014%, 0.034%, and 0.014% for *E. coli*, *S. aureus,* and *S. enteritidis*, respectively (Figure 2). Otherwise, the optimal EO mixture for killing bacteria was composed of 50% ORE, 40% THY, and 10% LG, resulting in the MBC values of 0.021% for *E. coli,* 0.051% for *S. aureus,* and 0.036% for *S. enteritidis* (Figure 3). Therefore, a greater sensitivity was observed in *E. coli*, followed by *S. enteritidis* and *S. aureus* for the MIC and MBC values. The two optimal theoretical proportions were further validated experimentally (Table 3). No difference was observed for the MIC and MBC values of *E. coli*, *S. aureus*, and *S. enteritidis* between theoretical values from desirability function and experimental values (Table 3).

## 3. Discussion

It is worth highlighting that this study is the first report investigating the antimicrobial potential of these three combined EOs against *E. coli*, *S. aureus*, and *S. enteritidis* using the mixture design. Moreover, this study aimed to enhance the antibacterial effect through decreased MIC and MBC values (dependent variables), and the lower or negative coefficient values demonstrate that the independent variables (single EO and their mixtures) were effective for increasing the antibacterial activity.

It is well-known that the antimicrobial activity of EOs is attributed to their composition and biochemistry, wherein the lipophilic nature of hydrocarbon skeletons and the hydrophilic nature of the other functional groups play an essential role in this activity [34]. The oregano and thyme EOs showed a high concentration of phenols, followed by hydrocarbons. The oregano EO was also composed of other functional groups, such as alcohol, ether, and ketone, with the thyme of alcohol and ether. Regarding lemongrass, its composition was rich in aldehydes, followed by alcohol, ester, and ketone (Section 2.1).

The single antimicrobial activity of oregano, thyme, and lemongrass EOs has been well-reported in the literature. Previous studies observed bacteriostatic and bactericidal activities of oregano against strains of *E. coli*, *S. enteritidis*, and *S. aureus* [35,36,37]. However, in our study, single ORE was only effective in killing *S. aureus*. This activity observed in the ORE^1^ can be justified by the greater bactericidal effect of the carvacrol against Gram-positive than Gram-negative bacteria [38].

Regarding THY, compounds such as thymol, *p*-cymene, *o*-cymene, γ -terpinene, and linalool are known to be responsible for the broad antibacterial spectrum of this EO [39,40]. The bacteriostatic and bactericidal effects of the thyme EO against *E. coli* [35], *S. enteritidis*, and *S. aureus* [37] have already been reported in the literature. The THY has similar antimicrobial properties to ORE since thymol is analogous to carvacrol [41], and it also possesses the same outer membrane disintegration properties, in addition to affecting a variety of cellular functions [42]. Furthermore, the predominance of carvacrol or thymol associated with the high concentration of hydrocarbons (greater hydrophobicity) may explain our findings concerning the similar activity of both ORE and THY against Gram-negative and Gram-positive bacteria.

The antibacterial activity of LG may be attributed to neral, geranial, and geraniol, among other compounds [43], and was observed by Naik et al. [44] against *E. coli* and *S. aureus*. However, there are no studies to date evaluating the MIC and MBC values of lemongrass against *S. enteritidis*. De Silva et al. [45] reported the bacteriostatic and bactericidal effects of lemongrass on *S. enterica*. Moreover, there is very little information in the literature about the mechanisms of action of neral and geranial compounds, but it is known that aldehydes are more active in Gram-positive bacteria than in Gram-negative ones [46], corroborating our findings (Table 4). In our study, LG^1^ showed higher MIC and MBC values (lower antimicrobial activity) than ORE^1^ and THY^1^ for all evaluated strains (Table 4).

Regarding the EO blends, the interaction between two EOs may be strain-dependent, which could explain our results for *S. aureus*. Furthermore, the similar composition of ORE and THY, with a predominance of phenols followed by hydrocarbons, may have impaired their combined activity since the synergistic or complementary effects are boosted by combining different functional groups [16,47]. According to Gallucci et al. [48], carvacrol and thymol had an antagonistic activity against a Gram-negative (*E. coli*) and a Gram-positive (*S. aureus*) bacteria, corroborating our findings concerning how the combination of ORE^0.5^ and THY^0.5^ has not been significant for the MIC and MBC values of any bacteria tested. On the other hand, according to Kalemba and Kunicka [34], the potential antimicrobial activity from EOs can be ranked in terms of chemical family as follows: phenols > aldehydes > ketones > alcohols > ethers > hydrocarbons, which reinforces our results concerning more antibacterial effectiveness for ORE^0.5^ + LG^0.5^ and THY^0.5^ + LG^0.5^ (Table 2). The combination of phenols (ORE and THY) and aldehydes (LG) are considered the most active EO functional groups against the bacteria [34,49]. Along with this, the hydrocarbon compounds present in ORE and THY can increase the activity of the aldehydes into LG towards membrane permeability [42,50], allowing better antimicrobial activity. It is worth noting that the antibacterial activity by a combination of LG with ORE or THY has not yet been reported. Nevertheless, an additive and synergistic antibiofilm activity against a Gram-negative bacteria, *Cronobacter sakazakii* (CICC 21544), were observed combining citral with thymol and carvacrol, respectively [51]. The combination observed between aldehydes and phenols corresponds with the major compounds in LG, ORE, and THY (Section 2.1).

Outbreaks with *S. enteritidis*, *E. coli,* and *S. aureus* are very frequent, representing a public health concern in many countries worldwide [52,53,54,55], and EOs have been widely studied and reported as being promising and safe antibacterial agents [56]. That provide an alternative against these pathogens in meat, dairy, fruit, and vegetable products and even in drinking water [57,58]. Despite this, there are still neither studies evaluating the action of the combination of oregano, thyme, and lemongrass nor the simultaneous inhibition and inactivation of *S. enteritidis*, *E. coli,* and *S. aureus*. However, some authors also found successful mixtures of other EOs to bacterial and fungal inactivation using a simplex–centroid design. Chraibi et al. [27] observed a synergistic effect between *M. piperita* and *M. pulegium* EOs against *E. coli* (54%/46%), *S. aureus* (56%/44%), and *Candida tropicalis* (55%/45%), and attributed it to a synergy between alcohols and ketones. Likewise, Ouedrhiri et al. [16] reported that a combination of *Origanum compactum* (28%), *Origanum majorana* (30%), and *Thymus serpyllum* (42%) was effective in inactivating *Bacillus subtilis* and *S. aureus*, while for *E. coli* a combination of *O. compactum* (75%) and *O. majorana* (25%) was needed. These authors attributed the effectiveness of the blended EOs to the synergy between alcohols and phenols. The mixture optimization method is still underexplored for the identification and modulation of the antimicrobial activity of EOs. Moreover, the studies with this approach only aim to find the optimal EO ratios for each microorganism without intending simultaneous microbial inactivation.

## 4. Conclusions

Based on our findings, the antimicrobial efficacy of the studied blends depended on the contribution of each EO in the mixture and the target strains. The most effective EO blends in reducing the minimum inhibitory concentration (MIC), and the minimum bactericidal concentration (MBC) of the bacterial strains tested, were thyme combined with oregano or lemongrass. The ideal mixture for simultaneous inhibition of *S. enteritidis*, *E. coli*, and *S. aureus* was comprised of 75% oregano, 15% thyme, and 10% lemongrass, while inactivation was 50% oregano, 40% thyme, and 10% lemongrass. The EO blends obtained in the present study can be promising alternatives to chemical preservatives against foodborne pathogens besides being used with other technologies, such as modified-release encapsulation systems for food packaging. Furthermore, considering industrial applicability, multiresponse optimization from the desirability function would strongly contribute to ensuring food safety and minimizing public health risks concerning these potential foodborne pathogens simultaneously using natural and green technology.

## 5. Material and Methods

### 5.1. Plant Material and Selection of EOs for Study

The EOs of ginger (*Zingiber officinale*), eucalyptus (*Eucalyptus globulus*), oregano (*Origanum vulgare*), thyme (*Thymus vulgaris*), rosemary (*Rosmarinus officinalis*), and lemongrass (*Cymbopogon citratus*) were acquired commercially from Quinari^®^ (Ponta Grossa, PR, Brazil). All EOs were separately submitted to a preliminary test to assess their effectiveness in inhibiting a Gram-positive (*S. aureus*) and a Gram-negative (*E. coli*) bacterium (following Section 2.3). No EO had activity against *S. aureus*; however, oregano, thyme, and lemongrass EOs were the only ones to show inhibition against *E. coli,* and thus these EOs were selected for the present study (data not shown).

### 5.2. Characterization of the EOs

The composition of EOs was determined using gas chromatography (Agilent 7890A) coupled to mass spectrometry (GC-MS; 5975C mass detector) and an Agilent 7890A gas chromatograph equipped with a flame ionization detector (FID) following the analytical conditions described by Chagas et al. and de Oliveira [59,60]. A 5% diphenyl—95% dimethylpolysiloxane capillary column (DB-5 MS, 30 m × 0.25 mm × 0.25 µm) was used in both chromatography systems. In short, the oven temperature was programmed to rise from 60 °C to 240 °C at 3 °C/minute with helium at 1.0 mL/minute as carrier gas. Furthermore, 1.0 µL (EOs in hexane at 0.1%) was injected at 250 °C. The transfer line was kept at 260 °C, the ion source at 230 °C, and the analyzer at 150 °C. The mass detector was operated in electron ionization mode (70 eV), with 3.15 scans/second, and data were collected in the 40–350 m/z range. For quantification, the samples were injected at 280 °C, using the same column and analytical conditions described above, with hydrogen at 1.5 mL/minute.

### 5.3. Mixture Design and Statistical Analysis

An augmented simplex–centroid design was used to assess the effect of oregano (ORE), thyme (THY), and lemongrass (LG) EOs in antibacterial activity by Scheffé regression models [25,61,62]. Table 4 shows the experimental design with twelve runs, including three replications (experiments 7, 8, and 9) and additional points (experiments 10, 11, and 12).

The linear, quadratic, and special cubic least-squares regression models were subjected to analysis of variance (ANOVA) and found to understand the best fit of data. Then, the quality of the fitted models was verified based on *R*^2^, *R*^2^_adj._, and ANOVA. After this preliminary step, it was found that the data fit the quadratic model better. Thus, this model was used to obtain responses of the dependent variables (*Y*) in the independent ones (*X*) function, see Equation (1).
*Y* = α_1_*X*_1_ + α_2_*X*_2_ + α_3_*X*_3_ + α_12_*X*_12_ + α_13_*X*_13_ + α_23_*X*_23_ + ε(1)
where *Y* is the minimum inhibitory concentration (MIC) or minimum bactericidal concentration (MBC) against *E. coli*, *S. enteritidis*, and *S. aureus* expressed in % (*w*/*v*); α_1_, α_2_, α_3_ are the estimated parameters of the isolated EOs and α_12_, α_13_, α_23_ of the binary mixture; *X*_1_, *X*_2_, and *X*_3_ are the independent variables corresponding to the ratio of ORE, THY, and LG; and ε is an error term. The significance of the estimated coefficients was evaluated through ANOVA with Tukey’s post hoc test.

### 5.4. Antimicrobial Assays

#### 5.4.1. Microorganisms

In order to verify the potential effect of EOs on foodborne pathogens, *E. coli* ATCC 25922, *S. aureus* ATCC 14458, and *S. enteritidis* ATCC 13076 were used in this study. All microorganisms were obtained from the culture bank of the Oswaldo Cruz Foundation (FIOCRUZ, Rio de Janeiro, Brazil), and stored on nutrient agar (Kasvi, Italy) under refrigeration at the Center for Food Analysis (NAL) at the Federal University of Rio de Janeiro, where they were reactivated in 10 mL of brain heart infusion broth (BHI) (Kasvi, Spain) at 37 °C/18–24 h. After, strains were streaked on MacConkey agar (Kasvi, Spain), Baird Parker agar (Kasvi, Spain) supplemented with egg yolk tellurium (Sigma-Aldrich, Germany), and Xylose Lysine Deoxycholate agar (XLD) (Kasvi, Espanha) at 37 °C/18–24 h, respectively. Next, a characteristic colony of *E. coli*, *S. aureus*, and *S. enteritidis* were inoculated in individual tubes containing BHI broth and incubated at 37 °C/18–24 h for subsequent use in the assays described below.

#### 5.4.2. Determination of Minimum Inhibitory Concentration (MIC)

The MIC of EOs and their mixtures were executed through the microdilution method according to the Clinical and Laboratory Standards Institute [63]. For that, an aliquot of the strains in BHI broth was transferred for 5 mL of Mueller Hinton broth (KASVI, Madrid, Spain) and incubated at 37 °C until the turbidity McFarland standard of 0.5 (about 8 log CFU mL^−1^ of each bacterium). The samples were previously diluted in Tween 80 (0.8%; *w*/*v*). In this test, two-fold serial dilutions ranging from 16 to 0.00312% (*w*/*v*) were prepared in Mueller–Hinton broth in a 96-well U-bottom plate. Finally, 10 μL of the suspensions of each bacterium was inoculated into each well. After that, the microplates were then incubated at 37 °C for 24 h. The MIC was determined as the lowest EO concentration to prevent visible growth in each well. The control group was performed with Tween 80 without EO or their mixtures, and as expected, no interferences were observed in the concentration used (0.8% *w*/*v*).

#### 5.4.3. Determination of Minimum Bactericidal Concentration (MBC)

The MBC means the lowest concentration to kill 99.999% of bacteria cells [64]. An aliquot of 50 µL from negative wells (Section 5.4.2) was spread on Plate Count Agar (PCA) (NEOGEN, Heywood, United Kingdom) and then incubated at 37 °C for 24 h. The MBC value corresponded to the lowest concentration of EOs or their mixture when no colony was observed in the culture medium. The MIC and MBC analyses were carried out in triplicate.

#### 5.4.4. Statistical Analysis and Mixture Optimization

The MD approach, regression coefficients, ANOVA, and desirability function (D) were determined using DoE in the Statistica v.9.0 software (Stasoft, Tulsa, OK, USA) [25]. The D function was applied to obtain the optimum EO formulation against *E. coli*, *S. aureus*, and *S. enteritidis* simultaneously [26]. This methodology is based on transforming each response into a dimensionless scale of individual desirability (*d*_i_) where each response (*y*_1_, *y*_2_, …, *y*_m_) of the original set is transformed to a range from 0 ≤ *d*_i_ ≤1. The d_i_ are then combined using the geometric mean, which gives the overall desirability D (Equation (2)):(2)$$D=\sqrt[{m}]{d_{1}\times d_{2}\times \ldots d_{m}}$$
where *m* is the number of responses, and the simultaneous optimization process is reduced to the simple task of the variables’ level calculation that maximizes D. A specific response can be maximized, minimized, or assigned a target value. In this study, the minimum function was applied (Equation (3)).
(3)$$d=\left\{\begin{array}{c} \;\;\;\;\;\;\;\;\;\;\;1\;\;if\;y_{i}<L_{i}\;\;\;\;\;\;\;\;\;\;\;\;\;\;\;\;\;\;\;\;\;\;\;\;\;\;\;\\ {\left(\frac{U_{i}-y_{i}}{U_{i}-L_{i}}\right)}^{t}\;if\;L_{i}\leq y_{i}\leq U_{i}\;\;\;\;\;\;\;\;\;\;\;\;\;\\ \;\;\;\;\;\;\;\;\;\;\;\;0\;\;if\;y_{i}>U_{i}\;\;\;\;\;\;\;\;\;\;\;\;\;\;\;\;\;\;\;\;\;\;\;\;\;\;\; \end{array}\right.$$
where *U_i_* is the maximum acceptable value for a given response, *L_i_* is the lowest allowed value, *t* is a parameter that expresses the importance of *y_i_*, so that the individual desirability is closer to the minimum in the final result of optimization. The optimal conditions (critical points) were experimentally validated using Tukey and Levene’s test.

## Figures and Tables

**Figure 1 antibiotics-11-01572-f001:**
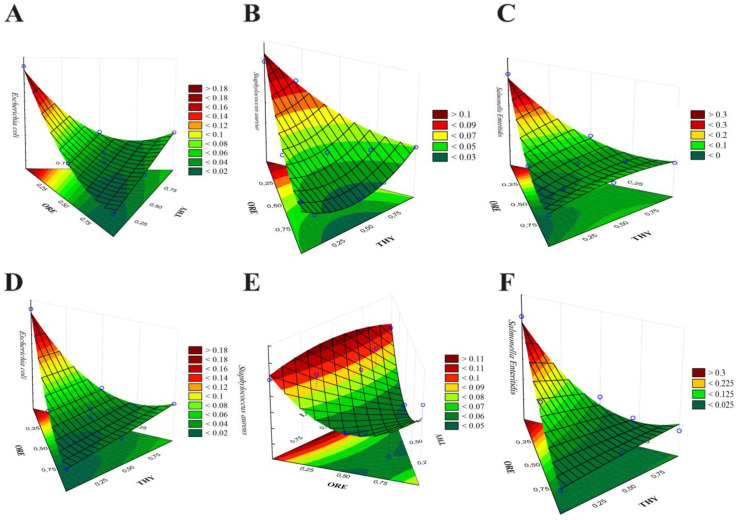
3D surface plots for the effect of different essential oil blends from oregano (ORE; *Origanum vulgare*), thyme (THY; *Thymus vulgaris*), and lemongrass (LG; *Cymbopogon citratus*) on minimum inhibitory concentration (MIC) value against *Escherichia coli* (**A**), *Staphylococcus aureus* (**B**) and *Salmonella* Enteritidis (**C**) and on minimum bactericidal concentration (MBC) value against *Escherichia coli* (**D**), *Staphylococcus aureus* (**E**) and *Salmonella enterica* serotype Enteritidis (**F**). Results are expressed in percentage (%) and are from twelve experiments, including three central replicates (Section 5.3).

**Figure 2 antibiotics-11-01572-f002:**
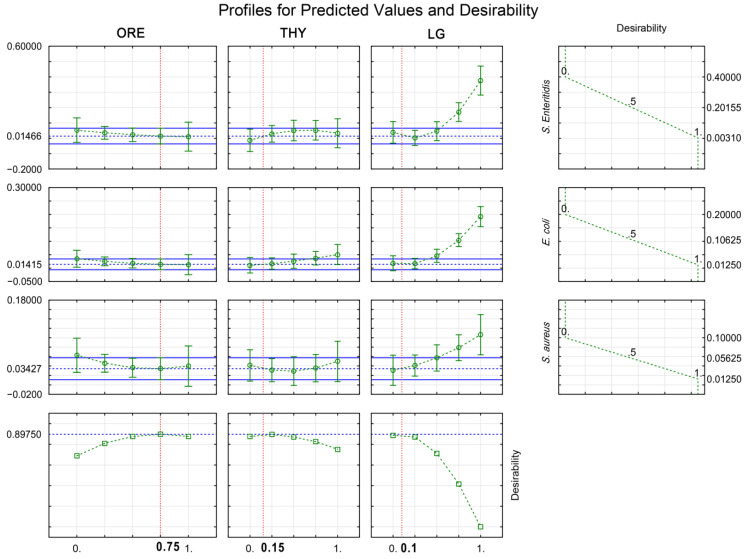
Desirability plot showing the optimal proportions of oregano (ORE; *Origanum vulgare*), thyme (THY; *Thymus vulgaris*), and lemongrass (LG; *Cymbopogon citratus*) to the simultaneous inhibition (MIC) of *Escherichia coli*, *Salmonella enterica* serotype Enteritidis, and *Staphylococcus aureus*. Results are expressed in percentage (%) and are from twelve experiments, including three central replicates (Section 5.3).

**Figure 3 antibiotics-11-01572-f003:**
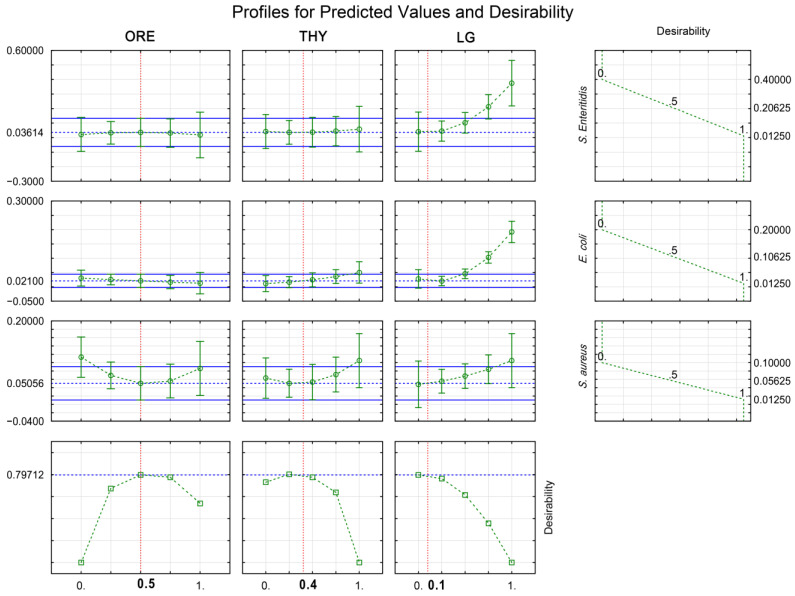
Desirability plot showing the optimal proportions of oregano (ORE; *Origanum vulgare*), thyme (THY; *Thymus vulgaris*), and lemongrass (LG; *Cymbopogon citratus*) to simultaneous inactivation (MBC) of *Escherichia coli*, *Salmonella enterica* serotype Enteritidis, and *Staphylococcus aureus*. Results are expressed in percentage (%) and are from twelve experiments, including three central replicates (Section 5.3).

**Table 1 antibiotics-11-01572-t001:** The quality of the quadratic model through analysis of variance (ANOVA) for the minimum inhibitory concentration (MIC) and minimum bactericidal concentration (MBC) experiments concerning essential oils (EOs) against *Escherichia coli*, *Staphylococcus aureus*, and *Salmonella enterica* serotype Enteritidis.

***E. coli* ^MIC^**	**Sum of Squares**	**Degrees of freedom**	**Mean Square**	***F*-value**	***p*-value**	** *R* ^2^ **	***R*^2^_adj._ ***
Model	0.026249	5	0.00525	20.83859	0.000995		
Total Error	0.001512	6	0.000252				
Lack of Fit	0.001095	4	0.000274	1.31386	0.475324	0.9456	0.9002
Pure Error	0.000417	2	0.000208				
Total Adjusted	0.02776	11	0.002524				
***S. aureus* ^MIC^**	**Sum of Squares**	**Degrees of freedom**	**Mean Square**	***F*-value**	***p*-value**	** *R* ^2^ **	***R*^2^_adj._ ***
Model	0.004934	5	0.000987	2.979	0.108212		
Total Error	0.001988	6	0.000331				
Lack of Fit	0.001987	4	0.000497	1490.163	0.000671	0.7128	0.4735
Pure Error	0.000001	2	0				
Total Adjusted	0.006921	11	0.000629				
***S. enteritidis* ^MIC^**	**Sum of Squares**	**Degrees of freedom**	**Mean Square**	***F*-value**	***p*-value**	** *R* ^2^ **	***R*^2^_adj._ ***
Model	0.112971	5	0.022594	14.03543	0.002927		
Total Error	0.009659	6	0.00161				
Lack of Fit	0.009242	4	0.002311	11.09053	0.084416	0.9212	0.8556
Pure Error	0.000417	2	0.000208				
Total Adjusted	0.12263	11	0.011148				
***E. coli* ^MBC^**	**Sum of Squares**	**Degrees of freedom**	**Mean Square**	***F*-value**	***p*-value**	** *R* ^2^ **	***R*^2^_adj._ ***
Model	0.026249	5	0.00525	20.83859	0.000995		
Total Error	0.001512	6	0.000252				
Lack of Fit	0.001095	4	0.000274	1.31386	0.475324	0.9456	0.9002
Pure Error	0.000417	2	0.000208				
Total Adjusted	0.02776	11	0.002524				
***S. aureus* ^MBC^**	**Sum of Squares**	**Degrees of freedom**	**Mean Square**	***F*-value**	***p*-value**	** *R* ^2^ **	***R*^2^_adj._ ***
Model	0.00622	5	0.001244	1.641339	0.280675		
Total Error	0.004548	6	0.000758				
Lack of Fit	0.002881	4	0.00072	0.864343	0.598648	0.5777	0.2257
Pure Error	0.001667	2	0.000833				
Total Adjusted	0.010768	11	0.000979				
***S. enteritidis* ^MBC^**	**Sum of Squares**	**Degrees of freedom**	**Mean Square**	***F*-value**	***p*-value**	** *R* ^2^ **	***R*^2^_adj._ ***
Model	0.106867	5	0.021373	4.81933	0.040813		
Total Error	0.02661	6	0.004435				
Lack of Fit	0.026193	4	0.006548	31.43148	0.031072	0.8006	0.6345
Pure Error	0.000417	2	0.000208				
Total Adjusted	0.133477	11	0.012134				

* R^2^_adj_: R^2^ adjusted. MIC: minimum inhibitory concentration; MBC: minimum bactericidal concentration.

**Table 2 antibiotics-11-01572-t002:** Coefficients of model fitted for minimum inhibitory concentration (MIC) and minimum bactericidal concentration (MBC) values concerning essential oils (EOs) against *Escherichia coli*, *Staphylococcus aureus*, and *Salmonella enterica* serotype Enteritidis and their level of significance.

	*E. coli*			*S. aureus*			*S. enteritidis*		
MIC	Estimation ^€^	SE ^†^	*p*-Value	Estimation	SE ^†^	*p*-Value	Estimation	SE ^†^	*p*-Value
ORE ^1^	0.012896	0.015265	0.430588	0.039735	0.017504	0.063661	0.011105	0.038586	0.783187
THY ^1^	0.050495	0.015265	0.016245 **	0.049711	0.017504	0.029566 *	0.033339	0.038586	0.420770
LG ^1^	0.192025	0.015265	0.000015 ***	0.106495	0.017504	0.000896 ***	0.376076	0.038586	0.000067 ***
ORE^0.5^ + THY^0.5^	−0.007719	0.068055	0.913390	−0.075322	0.078039	0.371723	0.172999	0.172033	0.353418
ORE^0.5^ + LG^0.5^	−0.225688	0.068055	0.016079 **	−0.060932	0.078039	0.464605	−0.645440	0.172033	0.009489 **
THY^0.5^ + LG^0.5^	−0.300477	0.068055	0.004494 **	−0.039770	0.078039	0.628516	−0.488530	0.172033	0.029577 *
	** *E. coli* **			** *S. aureus* **			** *S. enteritidis* **		
**MBC**	**Estimation ^€^**	**SE ^†^**	***p*-Value**	**Estimation**	**SE ^†^**	***p*-Value**	**Estimation**	**SE ^†^**	***p*-Value**
ORE ^1^	0.012896	0.015265	0.430588	0.086091	0.026477	0.017433 **	0.018346	0.064046	0.784173
THY ^1^	0.050495	0.015265	0.016245 **	0.104938	0.026477	0.007424 *	0.058587	0.064046	0.395587
LG ^1^	0.192025	0.015265	0.000015 ***	0.104938	0.026477	0.007424 *	0.374725	0.064046	0.001100 **
ORE^0.5^ + THY^0.5^	−0.007719	0.068055	0.913390	−0.188111	0.118046	0.162149	0.018773	0.285542	0.949716
ORE^0.5^ + LG^0.5^	−0.225688	0.068055	0.016079 **	−0.188111	0.118046	0.162149	−0.256087	0.285542	0.404348
THY^0.5^ + LG^0.5^	−0.300477	0.068055	0.004494**	0.051866	0.118046	0.675775	−0.622240	0.285542	0.072145

^†^ SE: standard error; * *p* < 0.05; ** *p* < 0.02; *** *p* < 0.001. ^1^ 100% of EO. ^0.5^ 50% of each essential EO. ^€^ Negative coefficient values indicate an increased antibacterial effect, and positive coefficient values suggest an antagonist effect.

**Table 3 antibiotics-11-01572-t003:** Observed values and validation for the optimal mixture of oregano (ORE; *Origanum vulgare*), thyme (THY; *Thymus vulgaris*), and lemongrass (LG; *Cymbopogon citratus*) essential oils (EOs) against *Escherichia coli*, *Staphylococcus aureus*, and *Salmonella enterica* serotype Enteritidis considering minimum inhibitory concentration (MIC) and minimum bactericidal concentration (MBC) values.

	MIC (%)_(*n* = 6)_ *				MBC (%)_(*n* = 6)_ *			
	Predicted Value	Observed Value	*t*-Test (%)	Levene’s Test (%)	Predicted Value	Observed Value	*t*-Test (%)	Levene’s Test (%)
*E. coli*	0.014	0.013	58.06	48.38	0.021	0.025	5.33	42.27
*S. aureus*	0.034	0.031	64.94	15.18	0.051	0.075	7.56	38.81
*S. enteritidis*	0.014	0.021	6.62	6.24	0.036	0.035	86.18	15.18

* The predicted proportions of EOs were 75% (ORE):15% (THY):10% (LG), and 50% (ORE):40% (THY):10% (LG) for MIC and MBC, respectively.

**Table 4 antibiotics-11-01572-t004:** Simplex–centroid design experiments, minimum inhibitory concentration (MIC) and minimum bactericidal concentration (MBC) values of single and blended essential oils (EOs) against *Escherichia coli*, *Staphylococcus aureus*, and *Salmonella enterica* serotype Enteritidis.

		Experiments *		MIC (%)			MBC (%)		
	ORE	THY	LG	*S. enteritidis*	*E. coli*	*S. aureus*	*S. enteritidis*	*E. coli*	*S. aureus*
1	1	0	0	0.0031	0.0125	0.05	0.0125	0.0125	0.1
2	0	1	0	0.025	0.05	0.05	0.025	0.05	0.1
3	0	0	1	0.4	0.2	0.1	0.4	0.2	0.1
4	0.5	0.5	0	0.05	0.025	0.025	0.05	0.025	0.05
5	0.5	0	0.5	0.05	0.05	0.05	0.2	0.05	0.05
6	0	0.5	0.5	0.1	0.05	0.05	0.1	0.05	0.1
7	0.33	0.33	0.33	0.05	0.05	0.05	0.05	0.05	0.05
8	0.33	0.33	0.33	0.025	0.025	0.05	0.025	0.025	0.05
9	0.33	0.33	0.33	0.05	0.025	0.025	0.05	0.025	0.1
10	0.67	0.17	0.17	0.025	0.0125	0.0125	0.025	0.0125	0.0125
11	0.17	0.67	0.17	0.05	0.025	0.05	0.1	0.025	0.1
12	0.17	0.17	0.67	0.05	0.05	0.1	0.05	0.05	0.1

The experiment was carried out in triplicate. * Proportion of each EO; ORE: oregano (*Origanum vulgare*) EO; THY: thyme (*Thymus vulgaris*) EO; LG: lemongrass (*Cymbopogon citratus*) EO.

## Data Availability

The data underlying this article will be shared on reasonable request to the corresponding author.

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
