# Peer review of "An Optimization of Oregano, Thyme, and Lemongrass Essential Oil Blend to Simultaneous Inactivation of Relevant Foodborne Pathogens by Simplex–Centroid Mixture Design"

_antibiotics, 2022, doi:10.3390/antibiotics11111572_

Round 1
Reviewer 1 Report
In the current paper, the simplex-centroid mixture design methodology coupled to a micro dilution assay to predict optimal essential oil blend to simultaneous inactivation of relevant foodborne. In my opinion, this manuscript needs some minor revisions before deciding about publication.
P1, L36: don’t use the abbreviation worlds.
In the introduction, use some the benefits of application of Mixture Designs in achieving optimized the natural essential oil or extract blends in food pathogens with references.
P2, L92-100: Rewrite this section. At first, the results of the analysis of the essential oils used in this research should be presented and it is recommended to show them in a table.
P3, L105: change “Tables 2 show” to “Table 2 shows”.
P9, the second paragraph: Explain the ideal ratios (critical values) of EOs to achieve the lowest MIC and MBC values for each bacteria with details.
Which pathogen was the most sensitive to the best ratio of essential oils?
P12, L94: Tables are not numbered sequentially. Change Table 1 to Table 4.
P13: Change Table 1 to Table 4.
P15, L58-63: write more details about the GC/MS analysis method such as carrier gas, oven program, … .
P15, L92: Where did microorganisms prepare from?
Author Response
November 02, 2022
Dear,
Please find attached a new version of our paper entitled “An optimization of oregano, thyme, and lemongrass essential oil blend to simultaneous inactivation of relevant foodborne pathogens by simplex-centroid mixture design” that was submitted to Antibiotics for publication.
The manuscript has been carefully rechecked, and appropriate changes have been made in accordance with the reviewers’ suggestions. We would like to thank the referees for their excellent comments. They have helped to improve the manuscript quality significantly, and we believe that it now provides a more balanced and better account of the research. We have modified the manuscript accordingly, and the detailed corrections are listed below.
Please do not hesitate to contact me again if further changes are required.
Sincerely,
Carlos Adam Conte-Junior, PhD.
Institute of Chemistry
Federal Universidty of Rio de Janeiro
Av. Athos da Silveira Ramos, 149, Cidade Universitária, Rio de Janeiro, RJ, Brazil
CEP: 21941-909
E-mail: conte@iq.ufrj.br
Tel: +55-21-3938-7825
Manuscript ID: antibiotics-1998752
Manuscript title: An optimization of oregano, thyme, and lemongrass essential oil blend to simultaneous inactivation of relevant foodborne pathogens by simplex-centroid mixture design.
Reviewer 1
Comments to the Author
In the current paper, the simplex-centroid mixture design methodology coupled to a micro dilution assay to predict optimal essential oil blend to simultaneous inactivation of relevant foodborne. In my opinion, this manuscript needs some minor revisions before deciding about publication.
Response: We are thankful for your revision. All comments were answered below.
P1, L36: don’t use the abbreviation worlds.
Response: Thank you for your observation. The keywords have been adjusted.
In the introduction, use some the benefits of application of Mixture Designs in achieving optimized the natural essential oil or extract blends in food pathogens with references.
Response: Thank you for your consideration. The benefits of the MD application were more clarified in the introduction (see lines 79-84).
P2, L92-100: Rewrite this section. At first, the results of the analysis of the essential oils used in this research should be presented and it is recommended to show them in a table.
Response: The table with the complete composition of essential oils have been added as supplementary material (see lines 94 and 95).
P3, L105: change “Tables 2 show” to “Table 2 shows”.
Response: Thank you for your observation. The adjustment has been made accordindly (line 107).
P9, the second paragraph: Explain the ideal ratios (critical values) of EOs to achieve the lowest MIC and MBC values for each bacteria with details.
Response: Thank you for your consideration. Adjustments were made to the text to clarify the ideal proportions of the EOs to achieve the lowest MIC and MBC values conforming suggested by the reviewer (please see page 9, lines 6-19).
Which pathogen was the most sensitive to the best ratio of essential oils?
Response: Thank you for the question. The validated model suggested different optimal proportions of EOs for each pathogen individually (Figure 1). However, only through in vitro evaluation with the optimal ratio of the three EOs against the three pathogens (section 2.3.) was proven that the most sensitive was E. coli with MIC and MBC values of 0.013% and 0.025%, respectively (Table 3). Due to reviewer’s comment, this information was added in the text (see page 10, lines 39-40).
P12, L94: Tables are not numbered sequentially. Change Table 1 to Table 4.
Response: Thank you for your observation. The table numbering has been adjusted (line 103).
P13: Change Table 1 to Table 4.
Response: The adjustment has been made (page 13).
P15, L58-63: write more details about the GC/MS analysis method such as carrier gas, oven program, … .
Response: According to the reviewer suggestion, more details on the methodology have been added (section 5.2.; lines 80-87).
P15, L92: Where did microorganisms prepare from?
Response: Thank you for your observation, more details on obtaining the bacteria were added to the methodology (see page 16, lines 118-121).
Please see the attachment.

Reviewer 2 Report
It was exceedingly hard to reconstruct what the authors had exactly done and how they had performed their experiments. They used two approaches, (Derringers) desirability function and augmented simplex-centroid design, without properly explaining these. They only wrote about "desirability function" and the reader is left to figure out that they meant Derringers desirability function. The augmented simplex-centroid design is only found 18 times in a Pubmed search with this search term, so one cannot assume that reader is familiar with it.
I found the presentation of the data very opaque. The reader sees lots of tables with statistical data, but it is very hard to discover the main message of the research from these tables. The figures are even worse explained and the units of the axes are often not given, nor explained, even though this is very much needed.
In conclusion, the research may be useful and the data valid, but that cannot be perceived from this manuscript.
Author Response
November 02, 2022
Dear,
Please find attached a new version of our paper entitled “An optimization of oregano, thyme, and lemongrass essential oil blend to simultaneous inactivation of relevant foodborne pathogens by simplex-centroid mixture design” that was submitted to Antibiotics for publication.
The manuscript has been carefully rechecked, and appropriate changes have been made in accordance with the reviewers’ suggestions. We would like to thank the referees for their excellent comments. They have helped to improve the manuscript quality significantly, and we believe that it now provides a more balanced and better account of the research. We have modified the manuscript accordingly, and the detailed corrections are listed below.
Please do not hesitate to contact me again if further changes are required.
Sincerely,
Carlos Adam Conte-Junior, PhD.
Institute of Chemistry
Federal Universidty of Rio de Janeiro
Av. Athos da Silveira Ramos, 149, Cidade Universitária, Rio de Janeiro, RJ, Brazil
CEP: 21941-909
E-mail: conte@iq.ufrj.br
Tel: +55-21-3938-7825
Manuscript ID: antibiotics-1998752
Manuscript title: An optimization of oregano, thyme, and lemongrass essential oil blend to simultaneous inactivation of relevant foodborne pathogens by simplex-centroid mixture design.
Reviewer comments
It was exceedingly hard to reconstruct what the authors had exactly done and how they had performed their experiments. They used two approaches, (Derringers) desirability function and augmented simplex-centroid design, without properly explaining these. They only wrote about "desirability function" and the reader is left to figure out that they meant Derringers desirability function. The augmented simplex-centroid design is only found 18 times in a Pubmed search with this search term, so one cannot assume that reader is familiar with it.
I found the presentation of the data very opaque. The reader sees lots of tables with statistical data, but it is very hard to discover the main message of the research from these tables. The figures are even worse explained and the units of the axes are often not given, nor explained, even though this is very much needed.
In conclusion, the research may be useful and the data valid, but that cannot be perceived from this manuscript.
Response: We are thankful for your consideration in making our manuscript more complete. The main objective of applying the simplex-centroid design and desirability function (D) to optimize the antimicrobial activity of EO blends is better detailed in the introduction (please see lines from 73 to 82). However, due to the reviewer’s comments, adjustments were made to the text to clarify it, and the D was better detailed in the methodology (please see section 5.4.4). The authors also indicated this section within section 2.3. (page 10, line 31) to facilitate the reading. The figure legends were better adjusted detailing the units, and the tables were better described concerning subtitles and title adjustments. In addition, a native speaker has revised the English language throughout the manuscript.
Please see the attachment.

Reviewer 3 Report
The manuscript is interesting and its findings are important for the food industry. It describes use of 3 essential extracts on foodborne pathogens. The methodology is sound, the results support their hypothesis and the references are up-to-date. Few minor corrections:
Lines 44-46: Please provide more details about the number of people affected.
Line 56: Also mention polyphenols and flavonoids as they also have antimicrobial properties
Line 103: Define MD
Table 2 legend: font size is not uniform
In Figure legends, define all abbreviations and give the number of biological replicates.
In Discussion, give few examples of appications of EOs in food and water safety
In Conclusions, suggest potential applications of these EOs in food security.
Conclusions should be after Discussion. Materials and Methods should be the last section.
Author Response
November 02, 2022
Dear,
Please find attached a new version of our paper entitled “An optimization of oregano, thyme, and lemongrass essential oil blend to simultaneous inactivation of relevant foodborne pathogens by simplex-centroid mixture design” that was submitted to Antibiotics for publication.
The manuscript has been carefully rechecked, and appropriate changes have been made in accordance with the reviewers’ suggestions. We would like to thank the referees for their excellent comments. They have helped to improve the manuscript quality significantly, and we believe that it now provides a more balanced and better account of the research. We have modified the manuscript accordingly, and the detailed corrections are listed below.
Please do not hesitate to contact me again if further changes are required.
Sincerely,
Carlos Adam Conte-Junior, PhD.
Institute of Chemistry
Federal Universidty of Rio de Janeiro
Av. Athos da Silveira Ramos, 149, Cidade Universitária, Rio de Janeiro, RJ, Brazil
CEP: 21941-909
E-mail: conte@iq.ufrj.br
Tel: +55-21-3938-7825
Manuscript ID: antibiotics-1998752
Manuscript title: An optimization of oregano, thyme, and lemongrass essential oil blend to simultaneous inactivation of relevant foodborne pathogens by simplex-centroid mixture design.
Reviewer comments
The manuscript is interesting and its findings are important for the food industry. It describes use of 3 essential extracts on foodborne pathogens. The methodology is sound, the results support their hypothesis and the references are up-to-date. Few minor corrections:
Response: We are thankful for your revision. All comments were answered below
Lines 44-46: Please provide more details about the number of people affected.
Response: Thank you for your observation. The numbers were included concerning the impact of foodborne diseases observed worldwide by the World Health Organization (WHO) (lines 42-44).
Line 56: Also mention polyphenols and flavonoids as they also have antimicrobial properties
Response: Thank you. Polyphenols and flavonoids were included in the text (please see lines 53 and 54).
Line 103: Define MD
Response: Thank you for your observation. The definition of MD was included (see line 103).
Table 2 legend: font size is not uniform
Response: Thank you. The adjustment has been made.
In Figure legends, define all abbreviations and give the number of biological replicates.
Response: Thank you for your observation. All figures’ legends were uploaded including the units, the number of experiments, and replicates conforming suggested by the reviewer.
In Discussion, give few examples of appications of EOs in food and water safety
Response: The examples have been added to the discussion (please see page 14, lines 25-27).
In Conclusions, suggest potential applications of these EOs in food security.
Response: Thank you for your suggestion. The potential application of the EO blends towards food security was included in the conclusion (please see page 14, lines 51-54).
Conclusions should be after Discussion. Materials and Methods should be the last section.
Response: Thank you for your observation. The sections have been moved (see page 14, line 43 and page 15, line 60).
Please see the attachment.

Reviewer 4 Report
S Ente should be introduced at th begining lie
Salmonella enterica serotype Enteritidis

Author Response
November 02, 2022
Dear,
Please find attached a new version of our paper entitled “An optimization of oregano, thyme, and lemongrass essential oil blend to simultaneous inactivation of relevant foodborne pathogens by simplex-centroid mixture design” that was submitted to Antibiotics for publication.
The manuscript has been carefully rechecked, and appropriate changes have been made in accordance with the reviewers’ suggestions. We would like to thank the referees for their excellent comments. They have helped to improve the manuscript quality significantly, and we believe that it now provides a more balanced and better account of the research. We have modified the manuscript accordingly, and the detailed corrections are listed below.
Please do not hesitate to contact me again if further changes are required.
Sincerely,
Carlos Adam Conte-Junior, PhD.
Institute of Chemistry
Federal Universidty of Rio de Janeiro
Av. Athos da Silveira Ramos, 149, Cidade Universitária, Rio de Janeiro, RJ, Brazil
CEP: 21941-909
E-mail: conte@iq.ufrj.br
Tel: +55-21-3938-7825
Manuscript ID: antibiotics-1998752
Manuscript title: An optimization of oregano, thyme, and lemongrass essential oil blend to simultaneous inactivation of relevant foodborne pathogens by simplex-centroid mixture design.
Reviewer comments
S Ente should be introduced at th begining lie
Salmonella enterica serotype Enteritidis
Response: We are thankful for your revision. The "Salmonella enterica serotype Enteritidis" was added at the first mention in the text (line 28) and in the figure and table titles. In the rest of the manuscript, we abbreviate it to S. Enteritidis.
Please see the attachment.

Round 2
Reviewer 2 Report
The figures were not changed, but a few sentences were added to the legends. I still do not understand the figures and my concerns about the tables have not been addressed. However, I see that the other reviewers do not share these concerns, so I think it is an editor's decision at this stage.